# Anti-Freezing Nanocomposite Organohydrogels with High Strength and Toughness

**DOI:** 10.3390/polym14183721

**Published:** 2022-09-06

**Authors:** Huijuan Zheng, Qiqi Huang, Meijun Lu, Jiaxin Fu, Zhen Liang, Tong Zhang, Di Wang, Chengpeng Li

**Affiliations:** School of Chemistry and Environmental Science, Guangdong Ocean University, Zhanjiang 524088, China

**Keywords:** nanocomposite, organohydrogels, anti-freezing, solvent replacement, high strength

## Abstract

Hydrogels based on nanocomposites (NC) structure have acquired a great deal of interest, but they are still limited by relatively low mechanical strength, inevitably losing elasticity when applied below subzero temperatures, due to the formation of ice crystallization. In this study, an anti-freezing and mechanically strong Laponite NC organohydrogel was prepared by a direct solvent replacement strategy of immersing Laponite NC pre-hydrogel into ethylene glycol (EG)/water mixture solution. In the organohydrogel, a part of water molecules was replaced by EG, which inhibited the formation of ice crystallization even at extremely low temperatures. In addition, the formation of hydrogen bonds between Laponite and the monomers of N-isopropylacrylamide (NIPAM) and hydroxyethyl acrylate (HEA) endowed the organohydrogels with high mechanical strength and toughness. The NC organohydrogel can maintain its mechanical flexibility even at −25 °C. The compressive stress, tensile stress, and elongation at the break of N_5_H_5_L reached 3871.71 kPa, 137.05 kPa, and 173.39%, respectively, which may be potentially applied as ocean probes in low temperature environment.

## 1. Introduction

Hydrogel is a polymer material with three-dimensional network structure [1,2,3,4] formed by physical or chemical crosslinking [5] of hydrophilic molecules. It is similar to biological tissues in terms of water richness and elasticity [6,7], which require a mild environment to maintain the basic properties of the gel [8]. However, almost water-based hydrogels are easy to freeze below subzero temperature [9,10,11], which makes the hydrogel change from soft material to hard material and lose its original toughness and strength. This hinders the application of hydrogels in cold environments [12]. Therefore, it is still a challenge to design hydrogel with tunable mechanical strength together with low temperature tolerance.

Currently, three approaches have been presented to develop hydrogels with low temperature tolerance. One is the introduction of inorganic salts into hydrogels, such as NaCl [13], LiCl, and CaCl_2_ [14]. The hydrogel synthesizes added inorganic salts or puts the hydrogel soaked in the salt water, dropping the ice point of water. Second strategy is preparation of the hydrogel in the organic compounds, such as ionic liquid [15,16,17,18,19]. This prepared hydrogel is a non-aqueous system, but the mechanical properties of the gel are weak due to the chemical cross-linking of the non-aqueous system. The last one is the introduction of a water-antifreeze agent binary solvent system, into hydrogel networks, through the one-step in-situ gelling or displacement [20,21,22]. Compared with the other two methods, water-antifreeze agent binary solvent can endow more stable mechanical properties. As a well-known antifreeze agent, ethylene glycol (EG) [23], glycerol (GL) [24], d-sorbitol (SB) [25], and DMSO [26] are suitable choices for fabricating organohydrogels and can inhibit the freezing of water as a result of forming strong hydrogen bonds with water molecules [27]. EG is a good organic solvent, which freezes at about −12 °C and is often used as an inhibitor of waterproofing freezing in industry. When using EG as the antifreeze agent in the organohydrogel networks, due to the large number of hydrogen bonds between EG and water molecules, the formation of an ice lattice was destroyed. As a result, the freezing point of EG and water mixture was greatly lower even than that of pure EG. Zhou [28] prepared organohydrogels by solvent replacement method. The water in the calcium-alginate/polyacrylamide (PAAm) hydrogel was replaced by the impregnation strategy to form an organohydrogel with EG-water binary solvent system. The elastic modulus of the organohydrogel was 40 kPa and the tensile strain reached 1700%. Yu [29] soaked MXene nanocomposite hydrogel (MNH) in an EG solution to replace part of the water molecules and develop an anti-freezing MXene nanocomposite organohydrogel. The optimal tensile strength and strain of the organohydrogel were 35 kPa and 1000%, respectively. Wu [30,31] prepared EG/GL-water antifreeze organohydrogels by a convenient solvent exchange method using the synthesized double-networks (DN) hydrogels as raw materials. The fabricated organohydrogel exhibited unprecedented tensile properties, its tensile strain was 1225%.

Nanocomposite (NC) hydrogels with good biocompatibility, non-toxicity, and better mechanical properties than traditional hydrogels were first reported in 2002 [32]. By introducing nano components [33] into the gel, hydrogels with high toughness and strength can be obtained [34,35,36]. It can be compressed, stretched, bent, and knotted without breaking, and its shape can be quickly restored when external forces are removed [37].

In this work, a tough NC organohydrogels with enhanced low temperature tolerance was fabricated by using an EG/water binary solvent system to displace water molecules in Poly(N-isopropylacrylamide)/poly(hydroxyethyl acrylate) (PNIPAM/PHEA)/Laponite hydrogel networks. Hydrogen bonds forming between Laponite, N-isopropylacrylamide (NIPAM), and hydroxyethyl acrylate (HEA) enhance mechanical properties of organohydrogels. EG disrupts the formation of ice crystal lattices of the residual water in gel, endowing extreme low temperature tolerance. Furthermore, tunable mechanical performance of the organohydrogels can be controlled by varying the monomer molar ratios of NIPAM and HEA. Therefore, NC organohydrogels, with enhanced and tunable mechanical properties, as well as extreme low temperature tolerance, could be designed and synthesized. These potentially expand practical applications of NC organohydrogels in low temperature environments.

## 2. Materials and Methods

### 2.1. Materials

LAPONITE XLG (Mg_5.34_Li_0.66_Si_8_O_20_(OH)_4_Na_0.66_, Laponite) was provided by Beijing Anwu Technology Co., LTD. (Beijing, China). Tetra sodium pyrophosphate (Na_4_P_2_O_7_) was obtained from Shanghai Kechang Fine Chemicals Co., LTD. (Shanghai, China). Tetramethylethylenediamine (TEMED), potassium peroxydisulfate (KPS) and hydroxyethyl acrylate (HEA) (96%) were purchased from Aladdin Co., LTD. (Shanghai, China). N-isopropylacrylamide (NIPAM, 99%) was provided by Beijing Bellingway Technology Co., LTD. (Beijing, China). Absolute ethanol (99.5%) from Guangdong Guanghua Sci-tech Co., LTD. (Guangdong, China) was used as solvent. All reagents were used as received without further purification. Deionized water was used in all experiments.

### 2.2. Preparation of Laponite NC Pre-Hydrogel

Laponite of 0.5 g was dispersed in 5 mL deionized water under stirring for 15 min to make a transparent dispersion. A desired amount of Na_4_P_2_O_7_ was added into the dispersion to reduce the viscosity. The mass ratio of Na_4_P_2_O_7_ to Laponite was kept at 0.0768:1 [38]. Then, a certain amount of monomer was added to the Laponite dispersion. After stirring for 15 min, 4 wt% KPS 500 μL and TEMED 10 μL were added in ice bath. The mixture was stirred for 5 min to allow complete dissolution of KPS. Finally, the dispersion was transferred to a sealed centrifugal tube, and the polymerization was carried out at 10 °C for 48 h. During the preparation, Laponite-based NC hydrogels with different monomer were obtained by adjusting the monomer molar ratio of NIPAM and HEA. The as-prepared hydrogels were cut into disks and soaked in a large amount of deionized water for five days. The deionized water was exchanged every day to remove monomers and oligomers which were not incorporated into the gels. The Laponite NC hydrogels dried in an oven at 80 °C until a constant weight were obtained.

### 2.3. Preparation of Laponite NC Organohydrogels

Laponite NC hydrogels with different monomer molar ratios prepared above were immersed in EG/water mixed solution, for 6 h, to obtain Laponite NC organohydrogels. Therein, the volume ratio of EG/water was fixed at 1:3, according to the reported procedure [30] but made some adjustment. The Laponite NC organohydrogels were denoted as NxHyL, where N, H, and L stood for monomers NIPAM, HEA, and Laponite, respectively, while x and y indicated the molar ratio of NIPAM/HEA was x:y. As a control, PNIPAM/Laponite (N_10_L) and PHEA/Laponite (H_10_L) were prepared according to the procedure above but in the absence of HEA and NIPAM, respectively. For example, N_5_H_5_L indicated the molar ratio of NIPAM/HEA was 5:5. In addition, in order to better distinguish the hydrogel before soaking, the Laponite NC hydrogels were named pre-hydrogel.

### 2.4. Characterization

The morphology and microstructure of pre-hydrogels were observed using a scanning electron microscope (SEM) (S-4800; Hitachi, Tokyo, Japan) and high resolution transmission electron microscope (HR-TEM) (jem2100; JEOL, Tokyo, Japan). For SEM, the accelerating voltage was 10 kV. All the samples were milled and stuck to the conductive glue. Particularly, the pre-hydrogel samples were frozen and dried in a freeze dryer (LGJ-10; Songyuan Huaxing, Beijing, China) in advance. For HR-TEM, the accelerating voltage was 200 kV. Ahead of TEM measurement, the oven-dried pre-hydrogel samples were milled and dispersed in acetone with ultra-sonication for 10 min. Then, a few drops of dispersions were dropped onto copper grids and dried at room temperature. Fourier transform infrared spectrometer (FTIR) (Nicolet 6700; Thermo Fisher Scientific, Massachusetts, USA) was used to record the vibration of functional groups of pre-hydrogels with the KBr pellets method in the wavenumber range of 4000 to 400 cm^−1^. The contents of monomers in pre-hydrogels were detected by X ray diffraction (XRD) (Ultima VI; Rigaku, Tokyo, Japan).

### 2.5. Mechanical Performance

The compressive tests were performed with the speed of 10 mm/min by UTM2203 electron compression tester and a load cell of 1000 N at room temperature of 25 °C. For compressive tests, both pre-hydrogels and Laponite NC organohydrogels with different monomer molar ratios were cut into a cylinder with a height of about 10 mm, and the compression stress was obtained at 90% strain. As for the cyclic compression tests, the compression stress was fixed at 60% strain with time intervals of 10 min. The tensile tests were performed on the pre-hydrogels and Laponite NC organohydrogels as a shape of strip with a size of 10 mm in width. The load cell was 100 N and the tensile speed was fixed at 50 mm/min. Differently, the strain for the cyclic tensile test was 200% and the time interval was 15 min. What is worth mentioning is that all cyclic tests were performed in triplicate, and glycerin was applied to the surface to avoid evaporation of water at 25 °C until the next measurement after each cycle.

### 2.6. Anti-Freezing Performance

The anti-freezing properties of Laponite NC organohydrogels were tested by taking out the organohydrogels from different low temperatures (0 °C, −5 °C, −15 °C, and −25 °C) and testing them at room temperature, 25 °C, immediately. The compressive measurement of organohydrogels which cut into a cylinder with a height of about 10 mm was carried at a speed of 10 mm/min. For tensile testing, the organohydrogels were cut into a strip shape with 10 mm in width, and the speed was 50 mm/min.

## 3. Results and Discussion

### 3.1. Preparation of Pre-Hydrogel and NC Organohydrogels

Figure 1 showed a general synthesis process to fabricate tough, anti-freezing Laponite NC organohydrogels. NIPAM/HEA was polymerized in Laponite dispersion to form pre-hydrogel. The interaction among hydrogel components will play an important role in the mechanical properties of the final NC hydrogel. Laponite, as a physical crosslinking agent, cross-linked with the monomer NIPAM and HEA to form a hydrogen bond, increased the intermolecular force of the organohydrogels, and improved the mechanical properties of the organohydrogels. Meanwhile, when pre-hydrogel was immersed in EG/water mixed solution, water molecules in the pre-hydrogel were replaced by EG. The binary solvent of EG and water displayed a low freezing point, resulting in the anti-freezing ability of organohydrogels.

### 3.2. Structure of the Pre-Hydrogels

In the polymerization process, with the increasing of polymerization time, the polymer chain inside the hydrogel was constantly extended and entangled. Finally, a three-dimensional network structure was formed. It was due to the hydrogen bond interaction between polymer chain and Laponite, which promoted the formation of three-dimensional network. In addition, with the increasing of the molar ratio of monomer HEA, the prepolymer became more viscous. After adding initiator KPS, the gelatin was formed quickly (Appendix A). The interaction between Laponite and monomer was studied by FTIR (Figure 2a), and the cross-linking between Laponite and monomer was confirmed. In the N_10_L spectrum, the Si-O tensile absorption peak of pure Laponite at 1012 cm^−1^ shifted to 1001 cm^−1^, attributing to the existence of a hydrogen bond between NIPAM and Laponite [39]. Similarly, the peak at 657 cm^−1^ shifted to the low peak of N_5_H_5_L at 656 cm^−1^, whereas it was at 668 cm^−1^ of N_10_L, indicating the hydrogen bond between NIPAM and HEA. In the XRD diffraction pattern (Figure 2b), the diffraction peak of pure Laponite at 19.6° was shifted to 20.0° [40] in N_10_L, and the diffraction angle of pure Laponite was shifted to 20.6° in N_5_H_5_L, while the interaction of NIPAM/HEA with Laponite was further confirmed. As seen from the HR-TEM images, Laponite acted as filamentous were uniformly distributed in the hydrogel and formed a homogeneous microstructure (Figure 2c,d). With increasing the molar ratio of NIPAM/HEA, a much stronger interaction between Laponite and NIPAM/HEA brought Laponite tightly intertwined with the polymer chain to form a more regular microstructure (Figure 2e,f).

### 3.3. Mechanical Properties of Pre-Hydrogels and NC Organohydrogels

In order to evaluate the mechanical properties of hydrogels, the compression and tensile properties of both pre-hydrogels and NC organohydrogels, with various molar ratios of NIPAM to HEA, were investigated (Figure 3a–d). Compared with conventional organic crosslinked hydrogels, the physical crosslinking of Laponite, NIPAM, and HEA improved the stiffness and toughness of the pre-hydrogels [41]. In the absence of NIPAM, the H_10_L hydrogel was very weak, and the compressive stress, tensile stress and elongation at break were only 571.91 kPa, 40.32 kPa, and 30.21%, respectively. With the increasing of the NIPAM to the molar ratio of NIPAM/HEA of 5:5, the compression and tensile stress of the pre-hydrogel reached the maximum of 2766.54 kPa (Figure 3a) and 153.59 kPa (Figure 3c), respectively. The elongation at break was 366.41%. Meanwhile, the compressive stress and tensile stress of pre-hydrogels, with various molar ratios of NIPAM/HEA at 1:9, 3:7, 5:5, 7:3, and 9:1, were higher than that of 10:0 and 0:10. Binary monomer (NIPAM and HEA) crosslinking with Laponite contributed to the improvement in the strength of pre-hydrogels [42,43]. That was similar to the literature of crosslinking with both SiO_2_ and TiO_2_ [44]. Furthermore, comparing the strength of pre-hydrogels of 10:0 and 0:10, the compression stress and tensile stress of pre-hydrogels with molar ratios of 10:0 were higher than that of 0:10. It suggested that NIPAM had a more leading effect on the mechanical property than HEA. Then, with the molar ratio of NIPAM/HEA went up from 5:5 to 10:0, the compressive stress and tensile stress started to decline. An explanation was due to the hydrogen bonds between NIPAM and HEA. It was speculated that, when NIPAM/HEA was 5:5, except for the NIPAM and HEA that interacted with Laponite, almost all the rest of NIPAM and HEA formed hydrogen bonds in the pre-hydrogel system. While other molar ratio of NIPAM/HEA, only part of NIPAM and HEA were combined together to form the hydrogen bonds. As a result, the maximum was at the molar ratio of NIPAM/HEA of 5:5. Nevertheless, the elongation at break was augmented, gradually, to 1097.31% as the molar ratio of NIPAM/HEA increased from 0:10 to 10:0. To further understand the variation of the mechanical property of the pre-hydrogels, the microstructures of the corresponding hydrogels were analyzed via SEM (Figure 3e–i). It was noticed that all of the pre-hydrogels showed typical honeycomb-like porous microstructures after being lyophilized, and the average pore sizes were about hundreds of micrometers. Meanwhile, a more uniform porous microstructure was shown in the molar ratios of 5:5. This could be proved the mechanical results above.

Analogously, the mechanical properties of the organohydrogel, after being soaked in EG/water mixture solution, also showed a similar trend of increasing first and then decreasing, and reached the maximum at the molar ratio of 5:5. The compressive stress, tensile stress, and elongation at break of N_5_H_5_L were 1695.58 kPa (Figure 3b), 117.67 kPa, and 356.17% (Figure 3d), respectively. Compared with the pre-hydrogels, it was found that the mechanical properties of organohydrogels decreased slightly. This may be caused by the swelling of the pre-hydrogel after immersion in EG/water mixture solution (Figure 3j).

Physical crosslinking can dissipate energy during the compressing and stretching process. Figure 4 showed the loading–unloading curves of N_5_H_5_L. No obvious hysteresis or energy loss was observed during the cyclic compressive process. The area of hysteresis loops, in cycles 2nd to 10th, was almost the same with the 1st cyclic loading. The organohydrogels displayed strong fatigue resistance. This could be attributed to the physical crosslinking of Laponite, which could dissipate energy effectively. In addition to the cyclic tensile tests, the hysteresis loop decreased slightly after the 3rd cycle. Upon loading, parts of the hydrogen bonds were broken to dissipate the energy, while the rest of the bonds maintained the macroscopic integrity of the organohydrogels. During the unloading process, the fractured hydrogen bonds rebuilt automatically with enough resting time [44,45]. It was the successful self-recovery of hydrogen bonds that made the organohydrogels excellent at fatigue resistance [46,47].

### 3.4. Anti-Freezing Properties of NC Organohydrogels

As for traditional hydrogels, when the ambient temperature drops below zero, hydrogels tend to freeze and lose their softness, deformability, and toughness, which greatly limits their application at low temperatures. In this study, organohydrogels with different monomer molar ratios were placed below zero of 0 °C, −5 °C, −15 °C, and −25 °C, respectively. However, it was found that, with the decrease in the molar ratio of NIPAM/HEA, the organohydrogels showed brittleness at −25 °C. As shown in the Figure 5a, under the pressure of gravity, the organohydrogels of N_10_L, N_9_H_1_L, N_7_H_3_L, and N_5_H_5_L showed good toughness, while N_3_H_7_L, N_1_H_9_L, and H_10_L were fractured. Combined with the mechanical properties of organohydrogels shown at Figure 3, N_5_H_5_L was selected to perform a series of mechanical test at low temperature. It displayed good elasticity at −25 °C and can recover quickly after compression (Figure 5b–d). Furthermore, N_5_H_5_L maintained high transparency and displayed good mechanical properties at −25 °C. It could be compressed, stretched, tied, twisted, and bent without any fracture (Figure 6 and Appendix A), whereas, when the pre-hydrogel was frozen at −25 °C, it changed solid white, brittle, and could be broken under the force. In conclusion, using EG to replace the water molecules in the pre-hydrogel, by immersing the pre-hydrogel in EG/water solution, can prevent the hydrogel from freezing at low temperature (Figure 7 and Appendix A). A strong hydrogen bond was formed between EG and water molecules [41,48], which suppressed the formation of ice lattice and reduced the freezing point of the hydrogel.

Organohydrogels also exhibited good anti-freezing properties. Compressive and tensile mechanical properties of organohydrogels, with different monomer molar ratios at low temperature, were measured. It can be seen from the Figure 8 that the organohydrogels still maintained good mechanical properties at −25 °C. The change trend of mechanical properties of organohydrogels for different monomer molar ratios at −25 °C was the same as the trend at 25 °C. It was still N_5_H_5_L that reached the maximum. Notably, the compressive stress of N_5_H_5_L at different low temperatures was always higher than that at 25 °C. This trend was consistent with the description in the literature [49]. Concomitantly, the compressive stress and tensile stress went up with the temperature, reduced from 0 °C to −25 °C. It can be found with the temperature dropped from 0 °C to −25 °C, the compressive stress and tensile stress increased from 2063.85 kPa and 119.07 kPa to 3871.71 kPa and 137.05 kPa, respectively. It can be explained that new ice crystals formed in the organohydrogel during low temperature and served as “reinforcing agents”, producing a hardening effect, which increased the rigidity of the organohydrogels [20,50]. Whereas, the elongation at break gradually decreased from 314.43% at 0 °C (Figure 8e) to 173.39% at −25 °C (Figure 8h) with the temperature went down. Although the ice crystals stiffen the gels, the gels were allowed to sustain large deformation through shear yielding between ice crystals. This was weakened ductility [46,51]. Besides, at the same temperature, the elongation at break gradually increased with the increasing of molar ratio of NIPAM/HEA. Compared to the elongation at break of N_10_L and H_10_L, the elongation at break of N_10_L was dramatically larger than that of H_10_L. This suggested that the presence of NIPAM had a more pronounced effect on the toughness of organohydrogels than HEA [32].

## 4. Conclusions

In summary, we successfully prepared Laponite NC organohydrogels by the solvent displacement method (using EG/water binary solvent system). The preparation method is simple and operated easily. The prepared organohydrogels have good antifreeze properties and can still maintain good mechanical properties even at −25 °C. The compressive stress of N_5_H_5_L reached 3871.71 kPa, and the tensile stress and elongation at break were 137.05 kPa and 173.39%, respectively. This is capable of adapting Laponite NC organohydrogels to variable lower temperatures. In addition, we expect that our findings could potentially expand practical applications as ocean probes in low temperature environment.

## Figures and Tables

**Figure 1 polymers-14-03721-f001:**
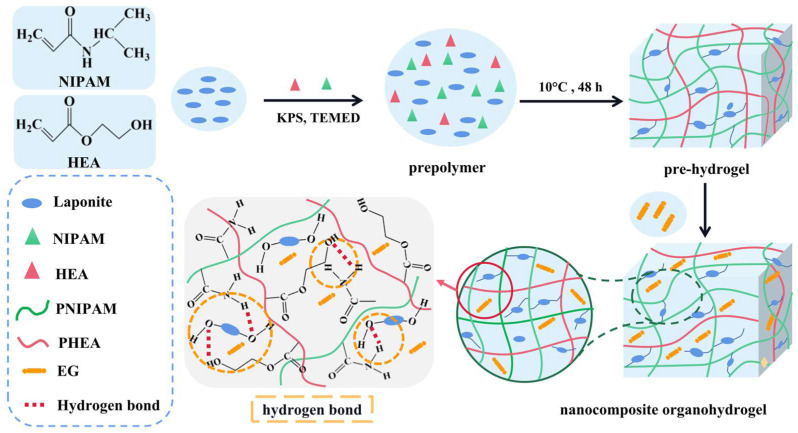
Schematic diagram of preparation of Laponite NC organohydrogels.

**Figure 2 polymers-14-03721-f002:**
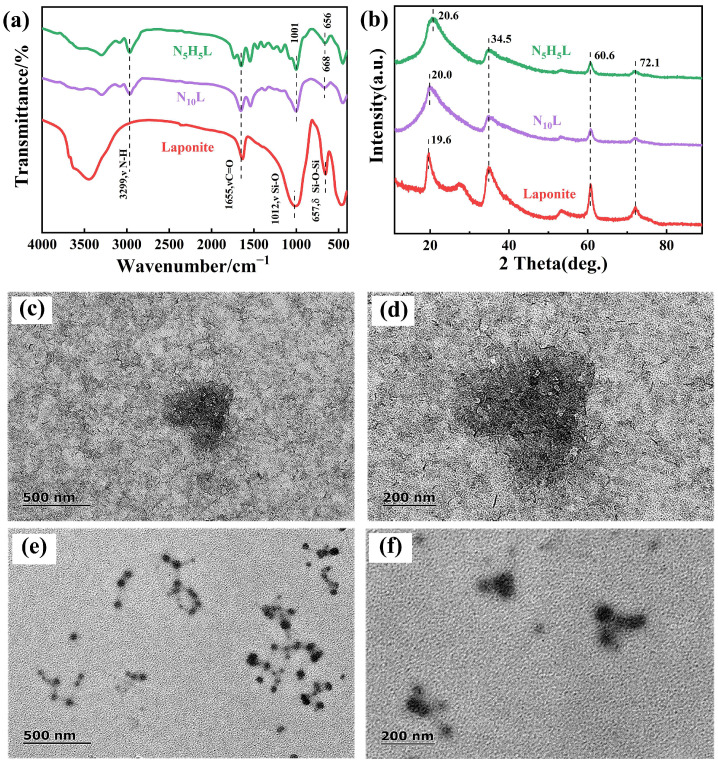
(**a**) FTIR spectra of pre-hydrogels; (**b**) XRD spectra of pre-hydrogels; (**c**,**d**) HR-TEM of the pre-hydrogels without HEA; (**e**,**f**) HR-TEM of the pre-hydrogels with molar ratio of NIPAM/HEA was 5:5.

**Figure 3 polymers-14-03721-f003:**
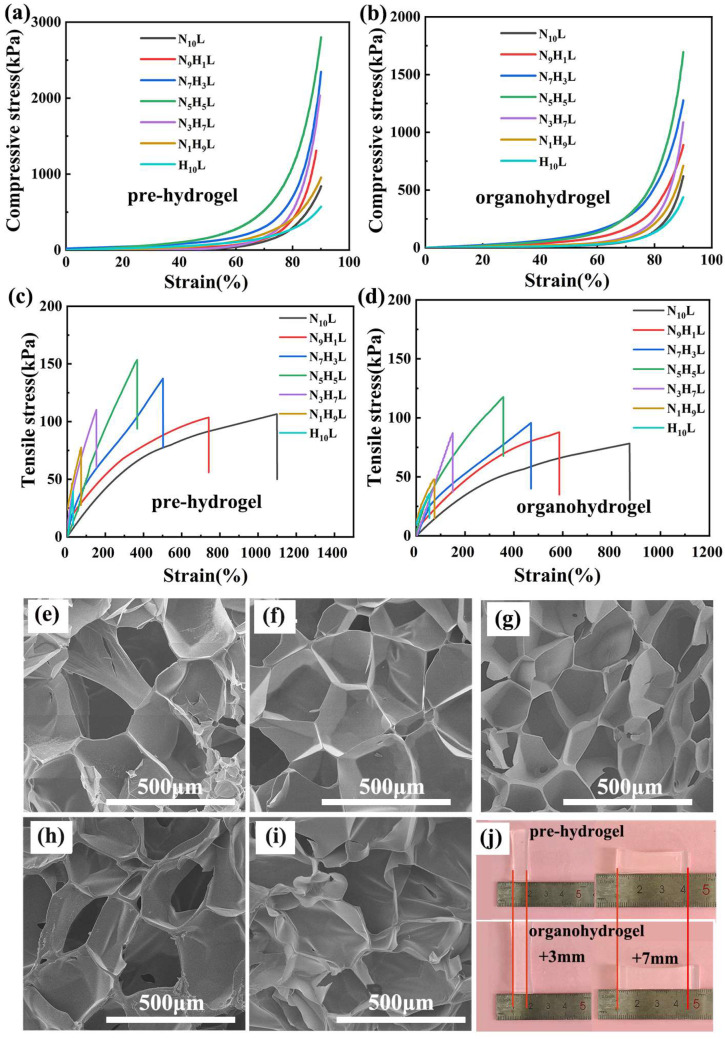
(**a**,**c**) Compression and tensile stress-strain diagram of pre-hydrogels; (**b**,**d**) Compression and tensile stress-strain diagram of NC organohydrogels; SEM of the pre-hydrogels with different molar ratios of NIPAM/HEA: (**e**) 10:0, (**f**) 7:3, (**g**) 5:5, (**h**) 3:7, (**i**) 0:10, respectively; (**j**) Comparison of hydrogels before and after being immersed in EG/water mixed solution.

**Figure 4 polymers-14-03721-f004:**
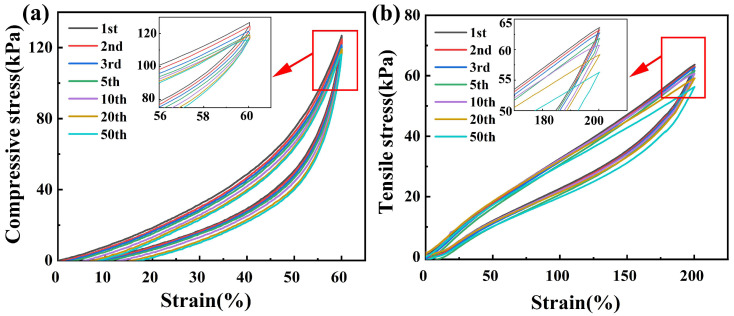
The loading-unloading tests of N_5_H_5_L. (**a**) cyclic compressive stress–strain curves at 60% strain for 50 cycles; (**b**) cyclic tension stress–strain curves at 200% strain for 50 cycles.

**Figure 5 polymers-14-03721-f005:**
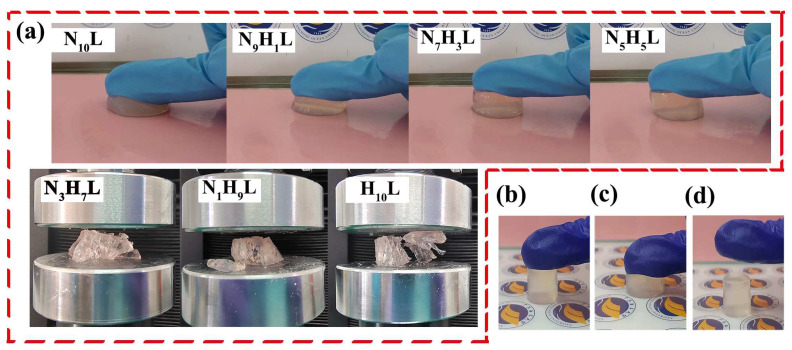
(**a**) Optical images of NC organohydrogel with different monomer molar ratios by pressing at −25 °C; (**b**–**d**) Optical compression images of N_5_H_5_L at −25 °C after 12 h. From left to right, it is before compression, during the compression and recovered after compression, respectively.

**Figure 6 polymers-14-03721-f006:**
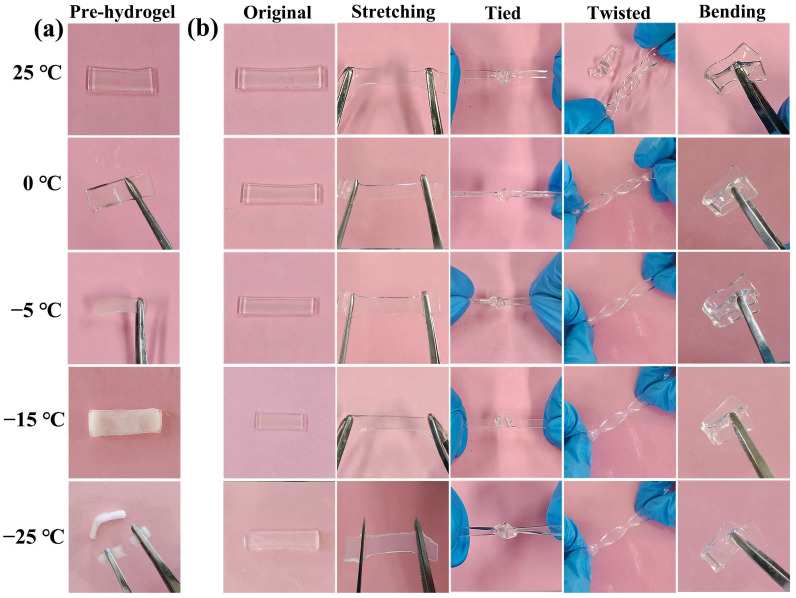
Optical images of hydrogels at 25 °C, 0 °C, −5 °C, −15 °C, −25 °C, respectively. (**a**) The pre-hydrogel with the monomer molar ratio of 5:5; (**b**) N_5_H_5_L organohydrogel of original, stretching, tied, twisted, and bending, respectively.

**Figure 7 polymers-14-03721-f007:**
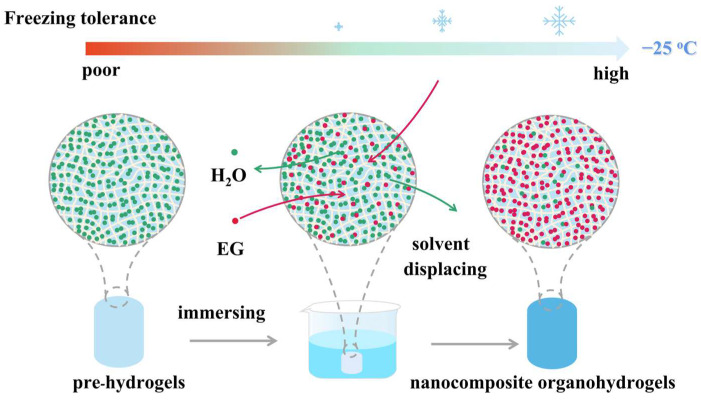
Schematic diagram of the anti-freezing properties of NC organohydrogels.

**Figure 8 polymers-14-03721-f008:**
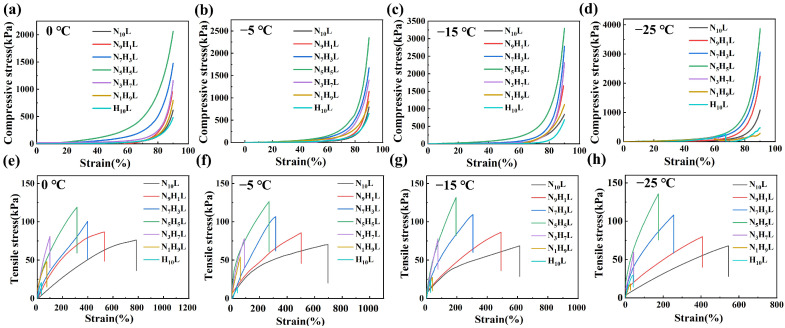
The compressive and tensile stress–strain diagrams of the NC organohydrogels after being placed at different low temperatures for 12 h and tested returned to room temperature: (**a**,**e**) 0 °C; (**b**,**f**) −5 °C; (**c**,**g**) −15 °C; (**d**,**h**) −25 °C; respectively.

## Data Availability

The data presented in this study are contained within the article.

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
