# Peer review of "Anti-Freezing Nanocomposite Organohydrogels with High Strength and Toughness"

_polymers, 2022, doi:10.3390/polym14183721_

Round 1
Reviewer 1 Report
The manuscript reports the synthesis of hydrogel immobilized with laponite and developed anti-freezing property by using solvent replacement strategy. In general, the research plan is a detailed and systematic and the conclusion is supported by the experiments. However, it appears to me that some parts of discussion were inadequate, and some clarifications are needed in order the results of this work to be persuasive. The suggestions are as follows
1. I suggest changing the following words for more appropriate meaning: Line 49 with --> which, Line 166 stretched --> extended, Line 247 destroyed --> suppressed.
2. Line 73-74: the abbreviations for HEA, PHEA are disordered. The abbreviation should be defined when it appears for the first time and consistently used later throughout the manuscript.
3. Line 95: why was the mass ratio of Na4P2O7:laponite fixed at 0.0768:1? Line 108: why was the volume ratio of EG/water fixed at 1:3? Are there any works reported about that before? If so, they must be cited.
4. The chemical structures of NIPAM and HEA should be added in Figure 1.
5. For the following statements in the discussion part, Line 200-202, Line 222, Line 224-225, and Line 275-278 should be compared with literature and cite references.
6. From SEM images in Figure 3, it did not clearly display the difference in pore size. In order to claim that the internal pores became smaller and the wall thickness change, Line 203-205, BET test should be tested to provide more informative results about porosity and pore size or size analysis is required. Also, the reason must be provided according to these changes with the variation of N-H-L.
7. Fig 3g and S1a, two squares appear behind the label letter.
8. The grammatical English expression of this manuscript / passive-active voice (e.g. Line 242-244) and typesetting (tie/twist labeled in Figure 6) need improvement.
9. Line 265-267, what did the authors mean “the compressive stress was always higher than that of 25C”?
10. Line 274: “Compared HEA”?
11. Please consider using other words instead of “What is more” in scientific report.
Reviewer 2 Report
The authors have explained in detail the synthesis of anti-freezing nanocomposite organohydrogels and evaluated their mechanical properties. They have used synthetic clay (Laponite) and incorporated ethylene glycol (EG) as the anti-freezing agent, which could disrupt water's ice crystal lattice formation. The mechanical strength of the hydrogel was enhanced by incorporating polymers capable of forming hydrogen bonds with the clay surface. The precursor for the hydrogel was made by using the polymerization of NIPAM/HEA in laponite dispersion. Various non-covalent interactions ensure the strong bonding of polymer with the laponite surface, which was proved by FT-IR and by analyzing the XRD pattern. Once the pre hydrogel is immersed in water, the EG molecules replace the water molecule, giving the system the anti-freezing property. However, its mechanical properties decreased slightly once the EG replaced the water molecule from the pre-hydrogel. The nanocomposite with a NIPAM/HEA -5: 5 proportion exhibited the best elasticity at -25° C among the various ratio tried.
· What could be the driving force for replacing water molecules from the pre-hydrogel in the presence of EG? How could authors quantify the amount of EG inside the hydrogel? Is there a complete replacement of water?
· How was the laponite dispersion prepared for the pre-hydrogel (before incorporating NIPAM/HEA)?
I recommend the publication of the present manuscript in the polymer journal once the authors address the aspects mentioned above.

Round 2
Reviewer 1 Report
-